# Structural equation modelling of the role of cognition in functional interference and treatment nonadherence among haemodialysis patients

**Frederick H. F. Chan**[1], **Pearl Sim**[1], **Phoebe X. H. Lim**[1], **Xiaoli Zhu**[1,2], **Jimmy Lee**[1,3], **Sabrina Haroon**[4], **Titus Wai Leong Lau**[4], **Allen Yan Lun Liu**[5], **Behram A. Khan**[6,7,8], **Jason C. J. Choo**[9,10], **Konstadina Griva**[1]*

1 Lee Kong Chian School of Medicine, Nanyang Technological University, Singapore, Singapore, 2 Nursing Services, National Healthcare Group Polyclinics, Singapore, Singapore, 3 Institute of Mental Health, Singapore, Singapore, 4 Division of Nephrology, Department of Medicine, National University Hospital, Singapore, Singapore, 5 Khoo Teck Puat Hospital, Singapore, Singapore, 6 Renal Health Services, Singapore, Singapore, 7 National University Health System, Singapore, Singapore, 8 Duke-NUS Medical School, Singapore, Singapore, 9 National Kidney Foundation, Singapore, Singapore, 10 Department of Renal Medicine, Singapore General Hospital, Singapore, Singapore

* konstadina.griva@ntu.edu.sg

**Data Availability Statement:** All relevant data are within the paper and its Supporting information files.

## Abstract

### Background and objectives

Cognitive impairment is common in haemodialysis patients and associated with adverse health outcomes. This may be due to cognitive impairments interfering with daily functioning and self-care, but evidence is limited. This cross-sectional study aims to explore the interrelationships between cognition and functional outcomes in haemodialysis patients.

### Methods

Haemodialysis patients completed measures of objective cognitive function (Montreal Cognitive Assessment), everyday problem-solving skills (scenario-based task), and subjective cognitive complaints (self-report). Participants also self-reported sociodemographic information, functional interference, treatment nonadherence, and mood and fatigue symptoms. Patients' clinical data including comorbidities and lab results were extracted from medical record. Structural equation modelling was performed.

### Results

A total of 268 haemodialysis patients (mean age = 59.87 years; 42.5% female) participated. The final model showed satisfactory fit: CFI = 0.916, TLI = 0.905, RMSEA = 0.033 (90% confidence interval 0.024 to 0.041), SRMR = 0.066, $\chi^2(493) = 618.573$ ($p < .001$). There was a negative association between objective cognitive function and subjective cognitive complaints. Cognitive complaints were positively associated with both functional interference and treatment nonadherence, whereas objective performance was not.

**Funding:** This work was supported by the Venerable Yen Pei-National Kidney Foundation Research Fund, Singapore [grant number NKFRC/2021/01/02]. KG received research funding from National Kidney Foundation Singapore. The funding sources had no role in the study design, recruitment of patients, data collection, analysis, interpretation of the results, writing of the manuscript, or decision to submit the manuscript for publication.

**Competing interests:** The authors have declared that no competing interests exist.

Everyday problem-solving skills emerged as a distinct aspect of cognition not associated with objective performance or subjective complaints, but had additive utility in predicting functional interference.

## Conclusions

Subjective cognitive complaints and everyday problem-solving skills appear to be stronger predictors of functional variables compared to objective performance based on traditional tests. Routine screening of everyday cognitive difficulties may allow for early identification of dialysis patients at risk of cognitive impairment, functional interference, treatment nonadherence, and poor clinical outcomes.

## Introduction

End-stage renal disease (ESRD) is the most advanced stage of chronic kidney disease where kidney function is irreversibly lost, necessitating dialysis or transplantation [1, 2]. Cognitive impairments (CIs) are common in ESRD patients receiving haemodialysis (HD) treatment, with more than 70% exhibiting at least mild impairments in one or more domains such as attention, memory, and executive function [3–6]. "Brain fog" has been a common complaint among dialysis patients [7] and a popular topic of discussion in online patient forums [8]. CIs in ESRD are associated with adverse health outcomes including dialysis withdrawal, hospitalisation, and mortality [3, 9–11]. These associations are assumed to be due to CIs interfering with daily functioning, decision-making, and self-management capabilities, however empirical evidence is scarce.

CIs are typically accompanied by functional interference because performance of everyday activities (e.g., personal hygiene, managing finances, etc.) is dependent upon the integrity of cognitive, motor, and sensory-perceptual skills [12]. In the context of ESRD, a specific aspect of daily functioning that is of particular relevance to patients is self-care and treatment adherence. HD patients are prescribed complex medical regimen that requires them to take daily medications, follow strict dietary guidelines, control fluid intake, and attend thrice-weekly dialysis sessions. These self-care activities are cognitively demanding and can be challenging for those with CIs. Medication taking, for instance, requires multiple cognitive processes including encoding and storage of health information (e.g., understanding the importance of taking medicine), executive function (e.g., developing a plan to adhere), prospective memory (e.g., remembering to take medicine on time), working memory (e.g., keeping the intention to take medicines active while preparing to take it), and source monitoring (e.g., remembering whether the medicine has been taken) [13].

Clearly cognition plays an essential role in HD patients' daily functioning and ability to maintain and optimise their health by following the prescribed medical regimen. However, currently we lack a comprehensive picture of the interplay between aspects of cognition and aspects of daily functioning in this population. Studies using neuropsychological tests, the gold standard measure of cognition, have found positive associations between objective cognitive ability and functional independence in dialysis patients [14]. Other studies have used self-reported measures to assess subjective cognitive complaints (SCCs). More frequent SCCs in HD patients have been found to be associated with greater functional impairment [15], and worse treatment adherence indicated by self-reports and laboratory results [16, 17]. Furthermore, scenario-based tasks that assess everyday problem-solving skills have been used. These

tasks require participants to generate solutions in response to problem scenarios, and are thought to reflect executive ability in real-world contexts [18]. Two studies found that everyday problem-solving skills positively predicted medication adherence in kidney transplant recipients [19, 20].

Despite these promising findings, a study that comprehensively examines the complex associations between cognition and real-world functioning in HD patients is lacking. As such, the current cross-sectional study aims to adopt structural equation modelling (SEM) analysis to disentangle the interrelationships among multiple aspects of cognition (i.e., objective performance, subjective complaint, and everyday problem-solving) and key outcomes (i.e., functional interference, treatment nonadherence, and clinical endpoints) in HD patients. SEM is a powerful and flexible statistical technique that integrates factor analysis, path analysis, and regression into a single framework [21]. It simultaneously accounts for multiple direct and indirect associations among a range of variables, hence allowing for the validation of complex theoretical models using a unified approach [21]. With this advanced statistical technique, this study will map key cognitive indicators of functional interference and nonadherence so that future interventions and support strategies could be developed to target the cognitive challenges that interfere with these key endpoints.

The hypothesised model to be tested in the current study is shown in Fig 1. We hypothesised that objective cognitive function and everyday problem-solving skills would covary (hence the bidirectional arrow) since they are both considered indicators of distinct aspects of cognitive abilities [18]. While objective tests reflect individuals' cognitive performance in optimal conditions, problem-solving tasks reflect individuals' problem-solving ability in everyday contexts. We also hypothesised that these two variables would directly contribute to SCCs since individuals with worse cognitive performance may experience more cognitive difficulties in daily lives, hence reporting more frequent SCCs [7, 22]. Furthermore, the three cognitive indicators were each hypothesised to have a direct effect on functional interference and treatment nonadherence based on previous evidence [14–17, 19, 20].

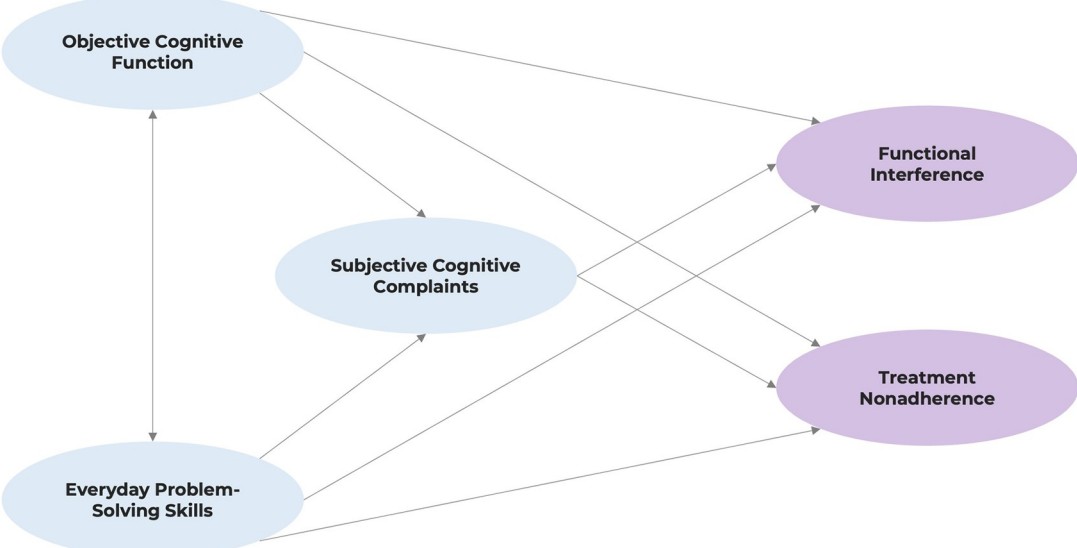

**Fig 1. Hypothesised model of associations between cognition, functional interference, and treatment nonadherence.**

## Materials and methods

### Participants

A convenience sample of HD patients was recruited from the National Kidney Foundation Singapore (NKF) between May 19 and November 4 2022. Ten NKF dialysis centres were selected to ensure geographical representation of dialysis centres across Singapore. The inclusion criteria were: (1) 21 years of age or older, (2) an estimated glomerular filtration rate lower than 15mL/min/1.73m$^2$, (3) having undergone HD treatment for at least 3 months, and (4) fluent in either English or Mandarin. The exclusion criteria were: (1) only fluent in dialects, (2) unable to give consent due to psychiatric diagnoses or established diagnosis of dementia, or (3) unable to complete survey due to visual or hearing impairments.

### Procedure

The study protocol was approved by the Institutional Review Board of the Nanyang Technological University (NTU-IRB-2021-025). A list of eligible patients was provided by the nurse managers of each dialysis centre. Study team members fluent in the patients' preferred language approached each patient during their dialysis sessions. After obtaining written consent, the following instruments were administered. Upon completion, patients were given a cash compensation.

### Measures

**Objective cognitive function.** The Montreal Cognitive Assessment (MoCA) was used to assess objective cognitive function [23]. The MoCA is a cognitive screening test that assesses visuospatial and executive functions (i.e., Trail-Making Test part B, cube copy, clock drawing, abstraction), attention (i.e., digit span forward and backward, vigilance, serial-7 subtraction), short-term memory (i.e., delayed recall), language (i.e., naming, sentence repetition, verbal fluency), and orientation (i.e., awareness of time and place) [23].

**Everyday problem-solving skills.** The Everyday Problem-Solving (EPS) task consists of real-world problem scenarios where participants were asked to generate solutions. Six scenarios that have been used in previously studies were used in the current study [19, 24]. Three scenarios described general daily problems and the other three described health-specific problems. For each scenario participants were asked to generate as many solutions as possible. The number of safe and effective solutions generated by each patient was used as an indicator of problem-solving skills [20].

**Subjective cognitive complaints.** SCCs were measured using the 33-item Patient's Assessment of Own Functioning Inventory (PAOFI) [25]. This measure assesses SCCs in four domains: memory (10 items), language (nine items), motor/sensory-perceptual ability (five items), and higher-level cognitive functions (nine items). Participants rated on a six-point Likert scale from "almost never" to "almost always" [25]. Mean scores of the four subscales were calculated, with a higher score indicating more frequent SCCs. Cronbach's alpha was 0.87 for the memory subscale, 0.89 for the language subscale, 0.73 for the motor/sensory-perceptual subscale, and 0.87 for the higher-level cognitive function subscale.

**Functional interference.** We assessed functional interference as a key dependent variable of the current study. The Work and Social Adjustment Scale (WSAS) is a measure of self-perceived functional interference in five domains (i.e., work, home management, social leisure activities, private leisure activities, and social relationships) attributable to an identified problem [26]. The original five items were used but preface was reworded to be cognition-specific.

Participants rated on a nine-point Likert scale ranging from 0 (i.e., "not at all impaired") to 8 (i.e., "very severely impaired"). Cronbach's alpha was 0.93 for WSAS.

**Treatment nonadherence.** Patients' self-reported medication nonadherence was measured by the five-item Medication Adherence Report Scale (MARS-5 ©Professor Rob Horne), which was rated on a five-point scale ranging from "never" to "always" [27]. A higher total score indicates poorer medication adherence. Cronbach's alpha was 0.77. The Dialysis Diet and Fluid non-adherence Questionnaire (DDFQ) was also assessed. DDFQ is a four-item scale that assesses frequency and degree of dietary and fluid nonadherence in dialysis patients [28]. Moreover, we collected patients' interdialytic weight gain (IDWG) as a clinical indicator of fluid adherence. Relative IDWG (i.e., the ratio of absolute IDWG to a patient's dry weight) was assessed prior to each dialysis session in the week of and the week before the survey date. The IDWG values were averaged across the two weeks. Additionally, the latest lab assay results (i.e., sodium [Na], potassium [K], calcium [Ca], phosphorus [PO4], and calcium-phosphorus product [Ca×PO4]) were collected and used as indicators of dietary and medication adherence.

**Mood and fatigue symptoms.** Patients' mood symptoms were measured by the two-item Patient Health Questionnaire (PHQ-2; $\alpha$ = 0.65) and the two-item Generalised Anxiety Disorder (GAD-2; $\alpha$ = 0.77). These two measures are brief screening tools of depression and anxiety that have been used in dialysis patients [29, 30]. Fatigue was measured using the one-item vitality subscale from the Kidney Disease Quality of Life questionnaire [31–33]. Higher scores indicate greater mood or fatigue symptoms.

**Sociodemographic and clinical information.** Self-reported demographic information was collected, including age, gender, ethnicity, education, relationship status, and employment status. Clinical information including primary kidney disease diagnosis, comorbidities, duration on HD, dialysis dose (Kt/V), and medication count, were extracted from patients' medical record.

## Statistical analyses

The steps of conducting the SEM analysis involved (1) testing of the baseline measurement model with confirmatory factor analysis (CFA), (2) specification of the structural model containing all hypothesised paths, and (3) modification of the structural model [34–36]. These steps were performed using the "lavaan" package with the WLSMV estimator in R 4.2.2 [37, 38]. To assess model fit, we used the Comparative Fit Index (CFI) [39], Tucker—Lewis Index (TLI) [40], the Root Mean Square Error of Approximation (RMSEA) [41], the Standardised Root Mean Square (SRMR) [21], and the Chi-square [21]. For CFI and TLI, values above 0.90 were considered to indicate adequate model fit, and values above 0.95 were considered to indicate excellent fit [39, 42–45]. RMSEA values lower than 0.06 and SRMR values lower than 0.08 were considered to indicate good fit [21].

## Model development

**Measurement model.** The measurement model was first constructed, with a total of eight latent variables, each measured by multiple indicator variables. Three latent variables were constructed to reflect different aspects of cognition, namely "objective cognitive function", "everyday problem-solving skills", and "subjective cognitive complaints". These three latent variables were indicated by the subscale/subdomain scores in the MoCA, EPS, and PAOFI, respectively. Two latent variables were constructed for the dependent variables, including "functional interference" measured by the five WSAS items, and "treatment nonadherence"

measured by self-report (i.e., MARS-5 and DDFQ), and physiological and biochemical parameters (i.e., IDWG, Na, K, Ca, PO4, and Ca×PO4).

Following previous SEM studies of cognitive function in patients with chronic disease [20, 34, 46], we constructed three additional latent variables in order to account for the confounding effects sociodemographic, clinical, and psychological factors in the hypothesised relationships. Specifically, a "sociodemographic" latent variable (indicated by age and years of education) and a "comorbidity" latent variable (indicated by presence of diabetes, hypertension, hyperlipidaemia, and cardiovascular disease) were constructed because these are established risk or protective factors of CIs in HD patients [47–49]. A "mood and fatigue symptoms" latent variable (indicated by PHQ-2, GAD-2, and KDQOL vitality item) was also specified because these symptoms can exacerbate SCCs [22] and are associated with behavioural and clinical outcomes in ESRD patients [50].

A CFA was first conducted to verify the measurement quality of the eight latent variables. The fit indices for the baseline measurement model were as follows: CFI = 0.839, TLI = 0.820, RMSEA = 0.043 (90% confidence interval 0.036 to 0.049), SRMR = 0.078, $\chi^2(566) = 810.166$ ($p < .001$), which suggested unsatisfactory model fit. All indicator variables had significant factor loadings on their corresponding latent variables except for two adherence indicators (i.e., Na and Ca). These two indicators were therefore removed from the model. However, fit indices for the revised measurement model were still in the unacceptable range, CFI = 0.853, TLI = 0.835, RMSEA = 0.043 (90% confidence interval 0.036 to 0.050), SRMR = 0.078, $\chi^2(499)$ = 719.322 ($p < .001$).

Modification indices suggested inclusion of error covariance between PO4 and Ca×PO4 in the model. These two variables were indeed highly correlated with each other, $r = 0.95$, $p < .001$. There is evidence that calcium and phosphorus are controlled by similar regulatory mechanisms [51], and high levels of serum phosphorus can combine with calcium to form calcium-phosphorus product [52]. It may therefore be theoretically justifiable to allow the residual terms of these two variables to freely covary [21]. We also added error covariance among DDFQ items due to the similar wordings of these questions, and we added error covariance between relative IDWG and the two fluid items in DDFQ as they are both thought to reflect fluid adherence. Following this modification, the final measurement model showed acceptable fit with the exception of Chi-square, CFI = 0.917, TLI = 0.905, RMSEA = 0.033 (90% confidence interval 0.024 to 0.041), SRMR = 0.065, $\chi^2(492)$ = 616.607 ($p < .001$). However, Chi-square test is sensitive to sample size with larger samples decreasing the $p$ value; the significant Chi-square was therefore not used as a basis for model rejection.

In the final measurement model, all factor loadings of the latent variables were at a significance level of $p < .05$. Detailed description of the final measurement model, including the latent and indicator variables, as well as their measurement, interpretation, mean values, percentage, and factor loadings, are presented in Table 1.

**Structural model.** A structural model was then constructed to examine the hypothesised regression paths between the latent variables specified in Fig 1. Objective cognitive function and EPS skills were hypothesised to covary with each other, and to have negative effects on SCCs. These three cognitive indicators were also hypothesised to have direct effects on functional interference and treatment nonadherence. However, the regression path from objective cognitive function to treatment nonadherence was considered optional because, despite theoretical assumptions, extant empirical evidence in the context of kidney disease suggest no association between these two variables [19, 20].

In addition, regression paths were added from sociodemographic factors, comorbidity, and mood and fatigue symptoms, to each of the five latent variables in Fig 1 (i.e., objective cognitive function, everyday problem-solving skills, subjective cognitive complaints, functional

**Table 1. Description of latent variables and indicator variables of the final measurement model.**

| Latent Variable | Indicator Variable | Measurement/Example Item | Interpretation of Higher Values | Mean (SD) / N (%) | Factor Loading |
|---|---|---|---|---|---|
| Objective Cognitive Function | Visuospatial/ Executive | Trail-making test part B; cube copy; clock drawing; abstraction | Better visuospatial and executive functions | 4.07 (1.58) | 0.65 |
| | Attention | Digit span forward and backward; vigilance; serial-7 subtraction | Better attention ability | 4.76 (1.47) | 0.65 |
| | Memory | Delayed recall | Better memory | 2.34 (1.89) | 0.52 |
| | Language | Naming; sentence repetition; verbal fluency | Better language function | 4.38 (1.03) | 0.52 |
| | Orientation | Awareness of time and place | Better orientation | 5.71 (0.54) | 0.23 |
| Everyday Problem-Solving Skills | General Problem | "Now let's say that one evening you go to the refrigerator and you notice that it is not cold inside, but rather, it's warm. What would you do?" | Better problem-solving skills for general problems | 7.59 (3.91) | 0.75 |
| | Health Problem | "You accidentally took the wrong combination of medication. What do you do?" | Better problem-solving skills for health-related problems | 4.3 (2.27) | 0.84 |
| Subjective Cognitive Complaints | Memory | "How often do you lose things or have trouble remembering where they are?" | More frequent memory complaints | 2.16 (0.81) | 0.74 |
| | Language | "How often do you have difficulty thinking of the names of things?" | More frequent language complaints | 2.06 (0.86) | 0.81 |
| | Motor/Sensory-Perceptual | "How often do you have difficulty feeling things with your right hand?" | More frequent complaints about motor/sensory-perceptual abilities | 1.96 (0.9) | 0.71 |
| | Higher-Level Cognitive | "How often do you have difficulty finding your way about?" | More frequent complaints about higher-level cognitive functions | 1.74 (0.76) | 0.93 |
| Functional Interference | Work | "Because of my cognitive difficulties, my ability to work is impaired." | Greater impact of cognitive difficulties on work ability | 1.39 (1.96) | 0.80 |
| | Home Management | "Because of my cognitive difficulties, my home management (cleaning, tidying, shopping, cooking, looking after home or children, paying bills) is impaired." | Greater impact of cognitive difficulties on home management | 1.22 (1.96) | 0.85 |
| | Social Activities | "Because of my cognitive difficulties, my social leisure activities (with other people, such as parties, bars, clubs, outings, visits, dating, home entertainment) are impaired." | Greater impact of cognitive difficulties on social leisure activities | 1.33 (2.09) | 0.89 |
| | Leisure Activities | "Because of my cognitive difficulties, my private leisure activities (done alone, such as reading, gardening, collecting, sewing, walking alone) are impaired." | Greater impact of cognitive difficulties on private leisure activities | 1.23 (1.94) | 0.87 |
| | Relationships | "Because of my cognitive difficulties, my ability to form and maintain close relationships with others, including those I live with, is impaired." | Greater impact of cognitive difficulties on social relationships | 1.00 (1.69) | 0.81 |
| Treatment Nonadherence | MARS-5 | "I forget to take them." | Poorer medication adherence | 7.29 (2.80) | 0.60 |
| | DDFQ-1 | "How many days during the past 14 days didn't you follow your diet guidelines?" | More frequent diet nonadherence | 2.65 (3.93) | 0.33 |
| | DDFQ-2 | "To what degree did you deviate from your diet guidelines?" | Greater deviation from diet guidelines | 0.99 (0.93) | 0.59 |
| | DDFQ-3 | "How many days during the past 14 days didn't you follow your fluid guidelines?" | More frequent fluid nonadherence | 2.66 (3.99) | 0.41 |
| | DDFQ-4 | "To what degree did you deviate from your fluid guidelines?" | Greater deviation from fluid guidelines | 1.03 (0.95) | 0.64 |
| | K | Latest available laboratory results | Higher serum potassium | 4.76 (0.64) | 0.15 |
| | PO4 | | Higher serum phosphorus | 4.66 (1.17) | 0.20 |
| | Ca×PO4 | | Higher calcium-phosphorus product | 42.79 (11.36) | 0.19 |
| | IDWG | Relative interdialytic weight gain averaged across the week of and the week before the survey date | Higher relative interdialytic weight gain | 3.34 (1.06) | 0.13 |

*(Continued)*

**Table 1.** (Continued)

| Latent Variable | Indicator Variable | Measurement/Example Item | Interpretation of Higher Values | Mean (SD) / N (%) | Factor Loading |
|---|---|---|---|---|---|
| Sociodemographic | Age | Years of age | Older age | 59.87 (11.72) | 0.59 |
| | Education | Years of full-time education | Higher education level | 9.59 (3.56) | -0.61 |
| Comorbidity | Diabetes | Medical record | Presence of diabetes | 145 (54.3%) | 0.69 |
| | Hypertension | | Presence of hypertension | 232 (86.9%) | 0.58 |
| | Hyperlipidaemia | | Presence of hyperlipidaemia | 143 (53.6%) | 0.64 |
| | Cardiovascular disease | | Presence of cardiovascular disease | 141 (52.8%) | 0.47 |
| Mood & Fatigue Symptoms | Depression | "Over the last 2 weeks, how often have you been bothered by the following problems? Little interest or pleasure in doing things." | Higher depressive symptoms | 1.10 (1.46) | 0.82 |
| | Anxiety | "Over the last 2 weeks, how often have you been bothered by the following problems? Feeling nervous, anxious, or on edge." | Higher anxious symptoms | 1.06 (1.51) | 0.84 |
| | Fatigue | "How much of the time during the past 4 weeks did you have a lot of energy?" | Higher fatigue symptoms | 3.18 (1.45) | 0.43 |

Notes. *SD* = Standard deviation. *N* = Sample size. MARS = Medication Adherence Report Scale. DDFQ = Dialysis Diet and Fluid non-adherence Questionnaire.

K = Serum potassium. PO4 = Serum phosphorus. Ca×PO4 = Calcium-phosphorus product. IDWG = Interdialytic weight gain.

interference, treatment nonadherence), in order to account for their potential confounding effects [22, 47, 53–55]. The structural model was tested and respecified until a final model was determined by weighing model fit indices and model parsimony (i.e., a simpler model with fewer parameters is favoured over more complex models provided the models fit the data similarly well).

## Results

### Sample characteristics

We approached a total of 459 HD patients in NKF dialysis centres. Ninety patients were excluded due to ineligibility. Within the remaining 369 eligible patients, 268 consented to participate (response rate 72.6%). The main reasons for rejection were lack of interest and feeling unwell. Therefore, 268 patients were included in final analyses. Table 2 reports the sociodemographic and clinical profiles of the sample. The mean age of the sample was 59.87 (*SD* = 11.72). Patients on average had been on HD for 78.85 (*SD* = 62.80) months.

### Final structural model

The fit indices for the full structural model were as follows: CFI = 0.917, TLI = 0.905, RMSEA = 0.033 (90% confidence interval 0.024 to 0.041), SRMR = 0.065, $\chi^2(492) = 616.607$ ($p < .001$). The path between objective cognitive function and treatment nonadherence was indeed not statistically significant, consistent with prior work [19, 20]. We removed this path while retaining all other paths and reran the structural model, with fit indices as follows: CFI = 0.916, TLI = 0.905, RMSEA = 0.033 (90% confidence interval 0.024 to 0.041), SRMR = 0.066, $\chi^2(493) = 618.573$ ($p < .001$). A chi-squared difference test showed that this

**Table 2. Sample characteristics (N = 268).**

| | Mean (*SD*) / *N* (%) |
|---|---|
| Sociodemographic | |
| Gender | |
| Male | 154 (57.5%) |
| Female | 114 (42.5%) |
| Age (years) | 59.87 (11.72) |
| Range | 26–84 |
| Ethnicity | |
| Chinese | 151 (56.3%) |
| Malay | 80 (29.9%) |
| Indian or others | 37 (13.8%) |
| Years of education | 9.59 (3.56) |
| Relationship status | |
| In a relationship | 182 (67.9%) |
| Not in a relationship | 86 (32.1%) |
| Work status | |
| Working | 76 (28.5%) |
| Not working | 191 (71.5%) |
| Clinical | |
| Primary diagnosis | |
| Diabetic nephropathy | 122 (45.5%) |
| Glomerulonephritis | 49 (18.3%) |
| Hypertension | 36 (13.4%) |
| IgA nephropathy | 12 (4.5%) |
| Others/uncertain aetiology | 49 (18.3%) |
| Presence of diabetes | 145 (54.3%) |
| Presence of hypertension | 232 (86.9%) |
| Presence of hyperlipidaemia | 143 (53.6%) |
| Presence of cardiovascular disease | 141 (52.8%) |
| Duration on HD (months) | 78.85 (62.80) |
| Medication count | 12.76 (4.25) |
| Kt/V | 1.60 (0.24) |

Notes. *SD* = Standard deviation. *N* = Sample size. HD = Haemodialysis. Kt/V = Dialysis dose.

trimmed model, which was also the more parsimonious model, could explain the observed data equally well compared to the full structural model without a significant loss in data-model fit ($p = .101$).

 The final trimmed model is presented in Fig 2. For conciseness, only regression paths that were statistically significant were presented in the figure. The model revealed that objective cognitive performance negatively predicted SCCs, but was not associated with EPS skills, functional interference, or treatment nonadherence. SCCs were positively associated with both functional interference and treatment nonadherence, suggesting indirect effects of objective performance on these dependent variables through SCCs. On the other hand, EPS skills were found to be unrelated to either objective performance or SCCs, but had a negative association with functional interference, suggesting that individuals with worse EPS skills had more severe functional interference. Sociodemographic factors (i.e., age and years of education) were associated with objective cognitive function, EPS skills, and treatment nonadherence, whereas

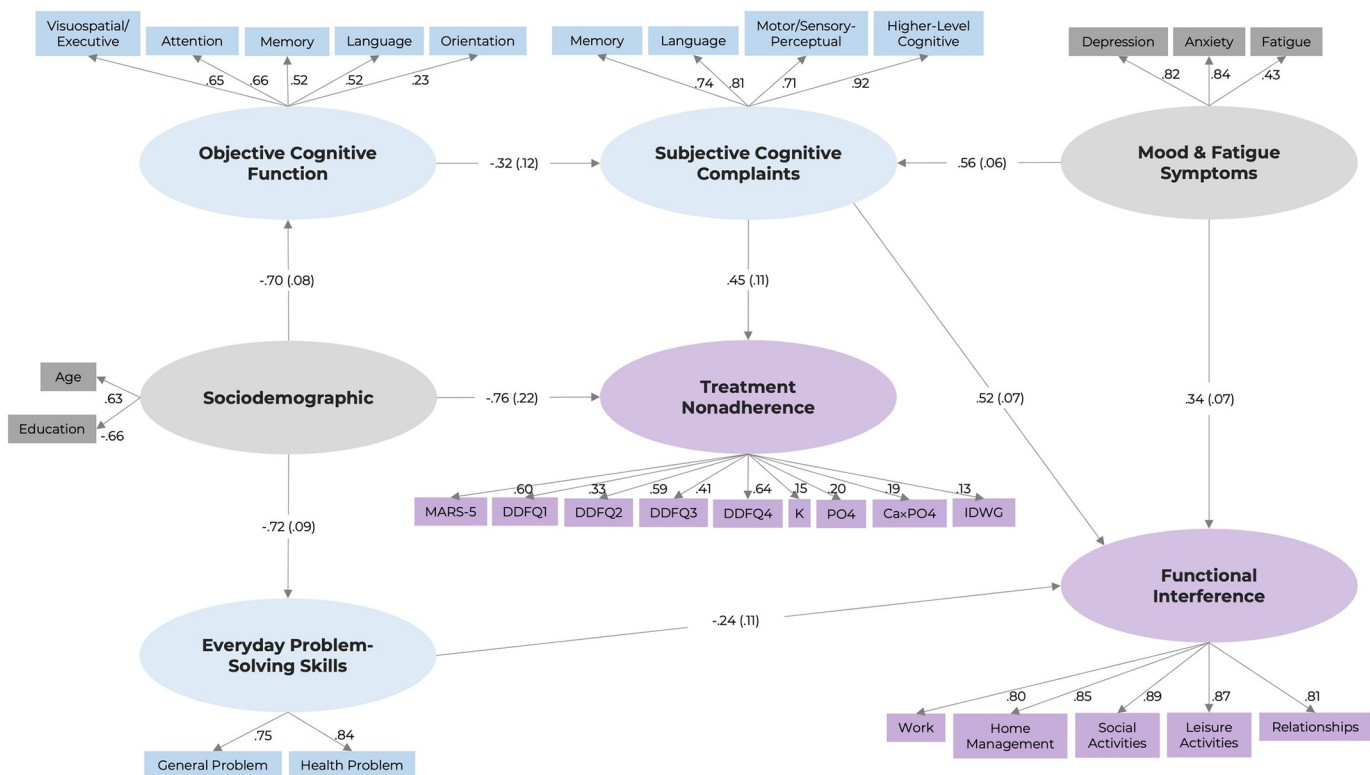

**Fig 2. Final structural model.** Standardised coefficients with standard errors are reported for regression paths. Standardised factor loadings are reported for paths between indicator variables and latent variables. Nonsignificant paths are not presented in the figure.

mood and fatigue symptoms were predictive of SCCs and functional interference. Comorbidity, however, was not associated with any other latent variables, and was therefore not presented in Fig 2. The final model explained 65.7% of the variance in functional interference (indicated by an $R^2$ of 0.657), and 63.3% of the variance in treatment nonadherence (indicated by an $R^2$ of 0.633).

## Discussion

To the best of our knowledge, this is the first SEM study to explore the complex interrelationships between multiple cognitive indicators and key functional and clinical variables in dialysis patients. The final model revealed interesting pathways through which cognition was directly and indirectly associated with functional interference and treatment nonadherence, highlighting cognitive difficulties as an important barrier in this population to functional capacity, life participation, treatment adherence, and disease management.

A key finding was that patients with more frequent SCCs also experienced more severe daily interference and exhibited poorer treatment adherence. In contrast, objective cognitive performance was only indirectly associated with these outcomes through SCCs. Indeed, it has been suggested that objective CIs based on neuropsychological tests do not necessarily translate to impairments in real-world functioning outside of test environment because some patients may adopt strategies to compensate for everyday cognitive failure, which is typically not allowed in standardised testing [56]. For some other patients, cognitive difficulties may be too mild to be detected by objective tests, but are nevertheless problematic for everyday life

[57]. Consistent with our findings, Song et al. found that SCCs, but not objective performance, predicted self-reported difficulties in performing activities of daily living in dialysis patients [15].

Importantly, treatment nonadherence in the present study was measured by both self-report and biochemical and physiological parameters. Inter-dialytic weight gain is typically thought to be an indicator of fluid adherence and sodium intake, whereas potassium and phosphorus may indicate dietary adherence [55]. Serum phosphorus can also indicate patient adherence to phosphate binders. Higher values of these clinical markers have been associated with poor survival [55]. Our findings are consistent with a recent longitudinal study where a reduction in SCCs over time was accompanied by a significant improvement in serum levels of PO4 and Ca×PO4 in HD patients [17]. Notably, the observed association between SCCs and nonadherence was significant even taking into account sociodemographic factors, comorbidity, and mood and fatigue symptoms, suggesting that this effect cannot be explained by other well-established determinants of nonadherence such as age and depression. Taken together, SCCs appear to be a useful measure that can simultaneously red-flag functional interference, nonadherence, and poor clinical outcomes in dialysis patients.

Another interesting finding was that patients with worse EPS skills were more likely to experience functional interference. The EPS task used in this study required participants to generate safe and effective solutions to the given scenarios. There is evidence in the cognitive development literature that the ability to generate a number of alternative solutions is a good indicator of problem-solving ability [58]. Patients who were unable to generate multiple solutions in response to the EPS task scenarios may also be more likely to experience failure in solving problems that arise in real-world contexts, which in the long term can impair independence in various aspects of life.

Unexpectedly though, EPS task performance was not associated with treatment nonadherence. This is inconsistent with two other studies where EPS skills were found to predict medication adherence in kidney transplantation recipients [19, 20]. These two studies, however, only assessed cognition using neuropsychological tests and the EPS task. It may be that SCCs have a stronger association with adherence and therefore by accounting for its effect in our model, the predictive value of EPS skills on adherence diminished. Indeed, the SCC measure used in our study is comprehensive and covers multiple cognitive domains, whereas EPS skills are dependent not just upon executive function, but also life experience, knowledge, and environmental factors [59].

We found that patients with worse cognitive performance on objective tests also subjectively reported more cognitive complaints. This observed association was modest, which is not unexpected based on previous work in other populations [22, 60]. Although objective tests and subjective reports both intend to capture the same underlying construct (i.e., cognition), they measure it using completely different methods and are influenced by different sets of predictors. While objective tests assess performance at a single time point in distraction-free environments, self-reports are based on accumulative daily experience that may be more reflective of longitudinal cognitive changes [60]. Objective and subjective cognition have therefore been considered as distinct constructs that complement each other with different utility in clinical and research settings. Assessing objective CIs can help establishing diagnosis of CIs and allow for advance care planning, whereas assessing SCCs can help identifying aspects of CIs that have the greatest impact on patients, hence allowing for a more patient-centred approach to managing this debilitating complication [61].

The current study has several important clinical implications. First, cognitive indicators that incorporate everyday scenarios (i.e., EPS and SCCs) appeared to be better predictors of real-world functional and clinical outcomes in HD patients, compared to traditional cognitive

tests. The associations of SCCs with underlying cognitive deficits and worse outcomes high-light the potential of SCCs as a stand-alone patient-reported outcome measure with clinical utility in dialysis settings that simultaneously signifies risks in multiple aspects of patients' health and well-being. Indeed, SCCs are increasingly considered as a prodromal marker of progression to dementia in Alzheimer's disease research [62, 63], and as a core patient-reported outcome in populations such as cancer [64] and HIV patients [65]. To date, research and clinical practice in the field of nephrology have been predominantly focusing on objective cognition, with SCCs being understudied and underrecognised. The emphasis on objective cognition is essential for diagnostic purposes, but hinders a comprehensive understanding of cognitive well-being in this population. Previous studies determined the utility of SCC measures solely based on their ability to distinguish patients with and without objective CIs [66]. We propose that SCCs should be treated as an outcome equally important as objective CIs, and should be assessed in combination with neuropsychological tests where possible.

Second, everyday cognitive difficulties reflected by SCCs and the EPS task emerged as potential modifiable risk factors for functional interference and treatment nonadherence. There is evidence that functional impairment is present in 21 to 85% of ESRD patients [11], and that optimal adherence is not achieved in as many as 80% of dialysis patients [55]. Our study showed that these high rates may in part be explained by the cognitive burden experienced by this population. It is therefore pivotal to develop and implement interventions that improve everyday cognitive abilities and compensate for cognitive lapses in this population so that the impacts of CIs on daily functioning and self-care can be mitigated. To date, there is a lack of research on the feasibility and effectiveness of cognitive interventions for ERSD patients. Future work in this area is needed to improve patient-centred care and to optimise adaptation to dialysis initiation.

Several limitations warrant acknowledgement. First, it should be noted that this is a cross-sectional observational study. Also, SEM is not inherently a causal method [21]. Caution is therefore needed when interpreting the significant paths in our final model. Although the directions of these associations have been hypothesised and tested, the study design and statistical method do not permit the interpretation of a cause-and-effect sequence. Future longitudinal studies and intervention research are needed to explore potential causal mechanisms and further validate our observations. Second, the MoCA used in this study is only a brief screening tool for global cognition. No comprehensive neuropsychological battery or standard diagnostic test was conducted to determine objective cognitive function. This is because the tools administered in this study were already very lengthy and required about 30–40 minutes for each patient to complete. A more comprehensive neuropsychological assessment was therefore not carried out considering patients' response burden and potential fatigue. Third, SCCs and functional interference were measured by self-report, which may be susceptible to recall bias. Future studies that incorporate informant-reports of SCCs and more objective measures of functional capacity may be needed. Finally, the study was conducted in a sample of HD patients, which limited the generalisability to other kidney disease subgroups such as peritoneal dialysis and kidney transplantation.

## Conclusion

In summary, this study adopted the SEM technique to disentangle the complex associations between cognition and key functional and clinical parameters in HD patients. Results revealed the central role of SCCs in indicating not just underlying cognitive deficits but also functional interference, treatment nonadherence, and suboptimal clinical outcomes. In contrast, EPS skills were found to be associated with only functional interference but not nonadherence,

whereas performance on traditional tests did not associate with any of these variables. It may be important to screen for SCCs in dialysis patients which would allow for early identification of the high-risk population, and subsequently early prevention or intervention strategies to mitigate the consequences of CIs.

## Supporting information

**S1 Data.**
(CSV)

## Acknowledgments

We thank the nurse managers and staff of the dialysis centres of National Kidney Foundation Singapore for facilitating the patient recruitment procedures and providing patients' medical data. We would also like to thank Kevin Tan, Shane Tan, Sze Ing Tan, and Nicole Tan for their assistance in patient recruitment and data entry.

## Author Contributions

**Conceptualization:** Konstadina Griva.

**Data curation:** Frederick H. F. Chan, Pearl Sim, Phoebe X. H. Lim, Behram A. Khan.

**Formal analysis:** Frederick H. F. Chan, Konstadina Griva.

**Funding acquisition:** Jimmy Lee, Sabrina Haroon, Titus Wai Leong Lau, Behram A. Khan, Konstadina Griva.

**Investigation:** Frederick H. F. Chan.

**Methodology:** Frederick H. F. Chan, Jimmy Lee, Sabrina Haroon, Titus Wai Leong Lau, Behram A. Khan, Konstadina Griva.

**Project administration:** Frederick H. F. Chan, Pearl Sim, Phoebe X. H. Lim, Konstadina Griva.

**Supervision:** Konstadina Griva.

**Validation:** Frederick H. F. Chan.

**Visualization:** Frederick H. F. Chan.

**Writing – original draft:** Frederick H. F. Chan.

**Writing – review & editing:** Frederick H. F. Chan, Pearl Sim, Phoebe X. H. Lim, Xiaoli Zhu, Jimmy Lee, Sabrina Haroon, Titus Wai Leong Lau, Allen Yan Lun Liu, Behram A. Khan, Jason C. J. Choo, Konstadina Griva.

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
