## [Decision Letter · Decision Letter 0]

17 Sep 2024

PONE-D-24-17737Structural equation modelling of the role of cognition in functional interference and treatment nonadherence among haemodialysis patientsPLOS ONE

Dear Dr. Griva,

Thank you for submitting your manuscript to PLOS ONE. After careful consideration, we feel that it has merit but does not fully meet PLOS ONE’s publication criteria as it currently stands. Therefore, we invite you to submit a revised version of the manuscript that addresses the points raised during the review process.

We look forward to receiving your revised manuscript.

Kind regards,

Henry H.L. Wu, MBChB PGCert

Academic Editor

PLOS ONE

Additional Editor Comments:

Thank you for your submission. Please refer and address to the reviewer's comments

Reviewers' comments:

Reviewer's Responses to Questions

**Comments to the Author**

1. Is the manuscript technically sound, and do the data support the conclusions?

Reviewer #1: Yes

2. Has the statistical analysis been performed appropriately and rigorously? 

Reviewer #1: Yes

3. Have the authors made all data underlying the findings in their manuscript fully available?

Reviewer #1: Yes

4. Is the manuscript presented in an intelligible fashion and written in standard English?

Reviewer #1: Yes

5. Review Comments to the Author

Reviewer #1: This study investigates the relationships between subject and object cognitive complaints and functional interference and treatment nonadherence, emphasizing the importance of subjective cognitive evaluation in practical settings. The study addressed an important issue in hemodialysis patients with reliable results, The expression in the article is good and the content is relatively comprehensive.

As an important disease that is related to all-cause death and other adverse outcomes, cognitive impairment in HD patients has already been a research hotspot in clinical settings. The following are only some suggestions to the authors:

First, the authors applied the SEM model to demonstrate that subjective cognitive complaints and everyday problem-solving skills appear to be stronger predictors of functional outcomes compared to objective performance. However, as a cross-sectional study, it's hard to determine the cause-and-effect relationship between two or more variables, a kind of association in the results couldn’t easily be explained with one variable being the cause and the other one being the outcome. It is also not proper to regard functional interference and treatment nonadherence as a certain kind of outcome, which is also commonly used in a cohort study, so the expressions in the manuscript should be adjusted to avoid misunderstanding.

Also, the results of this study showed that subjective cognitive complaints were related to functional interference and treatment nonadherence, does that mean objective assessment of cognitive impairment is not necessary before the patients have complaints? Generally, we could have a valuable time window between objective and subjective cognitive impairment, which may provide effective interference to the patients.

Second, the MoCA used in this study is only a screening tool for mild cognitive impairments, the author should explain why they didn’t apply the standard diagnosis tests for cognitive impairments.

6. PLOS authors have the option to publish the peer review history of their article (what does this mean?). If published, this will include your full peer review and any attached files.

Reviewer #1: **Yes: **Yang Luo

---

## [Author Response · Author response to Decision Letter 0]

25 Sep 2024

We have attached a document within this revision and have replied to all reviewer comments. We thank the editor and reviewers for their positive feedback.

---

## [Decision Letter · Decision Letter 1]

30 Sep 2024

Structural equation modelling of the role of cognition in functional interference and treatment nonadherence among haemodialysis patients

PONE-D-24-17737R1

Dear Dr. Griva, 

We’re pleased to inform you that your manuscript has been judged scientifically suitable for publication and will be formally accepted for publication once it meets all outstanding technical requirements.

Kind regards,

Henry H.L. Wu, MBChB PGCert

Academic Editor

PLOS ONE

Additional Editor Comments (optional):

The authors have satisfied the comments from the initial review

Reviewers' comments:

Reviewer's Responses to Questions

**Comments to the Author**

1. If the authors have adequately addressed your comments raised in a previous round of review and you feel that this manuscript is now acceptable for publication, you may indicate that here to bypass the “Comments to the Author” section, enter your conflict of interest statement in the “Confidential to Editor” section, and submit your "Accept" recommendation.

Reviewer #1: All comments have been addressed

2. Is the manuscript technically sound, and do the data support the conclusions?

Reviewer #1: Yes

3. Has the statistical analysis been performed appropriately and rigorously? 

Reviewer #1: Yes

4. Have the authors made all data underlying the findings in their manuscript fully available?

Reviewer #1: Yes

5. Is the manuscript presented in an intelligible fashion and written in standard English?

Reviewer #1: Yes

6. Review Comments to the Author

Reviewer #1: No more comments to the authors. I think the authors have made enough change about the limitaion that they have after the revision. I recommend this paper to be published in your journal.

7. PLOS authors have the option to publish the peer review history of their article (what does this mean?). If published, this will include your full peer review and any attached files.

Reviewer #1: **Yes: **Yang Luo

---

## [Editor Report · Acceptance letter]

7 Oct 2024

PONE-D-24-17737R1 

PLOS ONE

Dear Dr. Griva, 

I'm pleased to inform you that your manuscript has been deemed suitable for publication in PLOS ONE. Congratulations! Your manuscript is now being handed over to our production team.

Kind regards, 

on behalf of

Dr. Henry H.L. Wu 

Academic Editor

PLOS ONE